# Counter-Acting *Candida albicans*-*Staphylococcus aureus* Mixed Biofilm on Titanium Implants Using Microbial Biosurfactants

**DOI:** 10.3390/polym13152420

**Published:** 2021-07-23

**Authors:** Erica Tambone, Alice Marchetti, Chiara Ceresa, Federico Piccoli, Adriano Anesi, Giandomenico Nollo, Iole Caola, Michela Bosetti, Letizia Fracchia, Paolo Ghensi, Francesco Tessarolo

**Affiliations:** 1Department of Industrial Engineering & BIOtech, University of Trento, 38123 Trento, Italy; erica.tambone@gmail.com (E.T.); giandomenico.nollo@unitn.it (G.N.); 2Department of Pharmaceutical Sciences, Università del Piemonte Orientale “A. Avogadro”, 28100 Novara, Italy; alice.marchetti@uniupo.it (A.M.); chiara.ceresa@uniupo.it (C.C.); michela.bosetti@uniupo.it (M.B.); letizia.fracchia@uniupo.it (L.F.); 3Department of Laboratory Medicine, Azienda Provinciale per i Servizi Sanitari, 38122 Trento, Italy; federico.piccoli@apss.tn.it (F.P.); adriano.anesi@apss.tn.it (A.A.); iole.caola@gmail.com (I.C.); 4Department CIBIO, University of Trento, 38123 Trento, Italy; dr.ghensi@gmail.com

**Keywords:** titanium coating, anti-biofilm coating, biosurfactants, mixed biofilm, *Staphylococcus aureus*, *Candida albicans*, dental implant, peri-implantitis, fungal-bacterial biofilm, cytotoxicity, scanning electron microscopy

## Abstract

This study aimed to grow a fungal-bacterial mixed biofilm on medical-grade titanium and assess the ability of the biosurfactant R89 (R89BS) coating to inhibit biofilm formation. Coated titanium discs (TDs) were obtained by physical absorption of R89BS. *Candida albicans*-*Staphylococcus aureus* biofilm on TDs was grown in Yeast Nitrogen Base, supplemented with dextrose and fetal bovine serum, renewing growth medium every 24 h and incubating at 37 °C under agitation. The anti-biofilm activity was evaluated by quantifying total biomass, microbial metabolic activity and microbial viability at 24, 48, and 72 h on coated and uncoated TDs. Scanning electron microscopy was used to evaluate biofilm architecture. R89BS cytotoxicity on human primary osteoblasts was assayed on solutions at concentrations from 0 to 200 μg/mL and using eluates from coated TDs. Mixed biofilm was significantly inhibited by R89BS coating, with similar effects on biofilm biomass, cell metabolic activity and cell viability. A biofilm inhibition >90% was observed at 24 h. A lower but significant inhibition was still present at 48 h of incubation. Viability tests on primary osteoblasts showed no cytotoxicity of coated TDs. R89BS coating was effective in reducing *C. albicans-S. aureus* mixed biofilm on titanium surfaces and is a promising strategy to prevent dental implants microbial colonization.

## 1. Introduction

The human oral microbiota includes over 700 microbial species that can interact with each other or with opportunistic pathogens, forming multi-species communities [1,2]. These communities persist on all biotic (mucous membranes, teeth) and abiotic (prostheses, dentures, implants) surfaces in the oral cavity, playing an essential role in oral health and diseases [3,4,5]. Among these pathologies with microbial etiology, peri-implantitis represents the main complication affecting patients with osseous-integrated dental implants, and can even lead to implant loss [6]. Peri-implantitis is defined as an inflammation of the soft tissues surrounding the implant, accompanied by a loss of supporting bone [7]. One of the main causes of the onset of peri-implantitis has been recognized in the microbial adhesion on the implant surface and subsequent biofilm formation [8,9,10].

Titanium and its medical-grade alloys, in particular, Ti6Al4V alloy, are the materials of choice for realizing a wide range of dental implant and implant trans-mucosal components [11]. This material has excellent biocompatibility and resistance to corrosion, also providing high strength resistance and overall good mechanical properties [12,13,14]. Once the titanium surface is exposed to the oral environment, it is quickly coated with a saliva film, rich in proteins. This conditioning film encourages the adhesion of microorganisms of the oral microbiota or the settling of microbial species introduced through the mouth, often resulting in the formation of polymicrobial biofilms [15].

Polymicrobial biofilms are a complex three-dimensional community composed of a heterogeneous group of microorganisms (bacteria, fungi, viruses) attached to biotic and abiotic surfaces, encapsulated in an extracellular polymeric matrix [16]. Within these multi-species communities, microorganisms are in close contact and interact in many ways to promote colonization and their survival. They often cooperate, resulting in increased pathogenicity, virulence and antimicrobial resistance, making infections difficult to manage, frequently requiring complex multi-drug strategies [17,18]. Recently, several studies were focused on fungal–bacterial interactions, given their relevance in human diseases [16,19,20,21].

*Candida albicans* is the most frequently isolated fungal species in the oral cavity [22]. Although it is a human commensal species, under certain conditions, such as the use of antibiotics, immunosuppressive treatments or immune-depressive diseases, and reduced salivation, it can cause infections of the oral mucosa [23,24,25]. In addition, due to its high capacity to form biofilms, it can colonize abiotic surfaces, such as prostheses and dental implants, being involved in the etiology of peri-implantitis [8,26,27]. Within the oral environment, *C. albicans* interacts with both oral bacteria [5,28,29] and opportunistic pathogens such as *Staphylococcus aureus* [30,31]. They form multi-species biofilms causing disorders such as caries, stomatitis, periodontal and peri-implant diseases. Although *S. aureus* does not belong to the oral microbiota, it is frequently introduced during implants surgery [32] and can early colonize implants and persists for a long time [33,34]. Several studies reported high levels of the bacterium on implants surface or in peri-implant pocket in patients affected by peri-implantitis [35,36].

Considering the key role of microbial biofilms in peri-implantitis and the peculiarities of fungal-bacterial mixed biofilm, finding new strategies to prevent implant colonization is a priority in dentistry. In recent years, different surfaces or coatings with antibacterial or anti-biofilm properties have been proposed to protect implants [37,38,39]. However, several limitations in safely and effectively implementing these strategies in clinical settings were reported [40]. Considering the multivariate causes of peri-implant disease, the right solution to prevent peri-implant diseases should identify strategies able to eliminate or minimize initial bacterial attachment without having a cytotoxic effect to allow and favor stable osseointegration and strengthen the peri-implant mucous seal [41].

Biosurfactants (BS) are natural molecules of microbial origin that, thanks to their low toxicity and high biocompatibility, have emerged as potential coating agents of medical devices to prevent their colonization. BS have shown interesting antibacterial, antifungal, antiviral and anti-adhesive properties that can be applied in medical and pharmaceutical fields as an alternative to conventional synthetic therapeutic agents [42,43,44,45]. Due to their amphiphilic structure and surface activity, BS can modify the chemical-physical properties of the surfaces, hindering microbial adhesion and biofilm formation on the devices [46,47]. In addition, these compounds can increase the permeability of microbial membranes, resulting in loss of metabolites and cell lysis, and alter proteins conformation, compromising vital functions, such as energy generation and transport [47,48].

Among BS, rhamnolipids are a class of glycolipids produced mainly by *Pseudomonas aeruginosa*, with multiple potential applications in pharmaceutical and biomedical fields [49,50,51], including oral-related health applications [52]. These molecules possess interesting antimicrobial and anti-biofilm properties, acting both as disrupting agents on pre-formed biofilms and as anti-adhesive agents limiting microbial adhesion on surfaces [53,54,55,56,57,58,59,60]. In most studies, biosurfactants’ activity was assessed against pure cultures of microorganisms and there is a lack of data in the literature concerning inhibition efficacy on mixed cultures.

In this paper, we aimed at defining a model for in vitro production of fungal-bacterial mixed biofilm on medical-grade titanium and exploiting this model to quantify the activity of a biopolymer coating made of rhamnolipids in inhibiting biofilm formation at the titanium surface for potential application as an anti-biofilm coating for dental implants.

## 2. Materials and Methods

### 2.1. Study Design

The study design was conceived considering the promising anti-biofilm results obtained on biosurfactant-coated titanium for inhibiting the formation of single *Staphylococcus spp.* biofilm [60]. The present study was composed of two main parts. In the first part, an in vitro model of single- and dual-species biofilm of *Candida albicans* and *Staphylococcus aureus* on titanium was developed. Single- and mixed dual-species biofilm of the two strains were grown on untreated titanium discs to evaluate the ability of the two strains to co-exist and form a mature and structured mixed biofilm. Biofilm growth and composition were quantified at 24, 48 and 72 h of incubation to assess their characteristics over time.

In the second part of the study, the efficacy of BS-coated titanium discs to counteract the formation of *C. albicans*-*S. aureus* dual-species biofilm was investigated. Experiments were performed comparing the biofilm growth on titanium discs coated with a 4 mg/mL BS solution to uncoated control discs, at 24, 48 and 72 h of incubation. Data were analyzed to provide the percentage of biofilm inhibition at the different time-points.

In both experimental parts, three complementary methods were used to quantify different biofilm characteristics: the crystal-violet (CV) method, was applied to measure the total biofilm biomass; the 3-(4,5-dimethylthiazolyl-2-yl)-2,5-diphenyltetrazolium bromide (MTT) reduction assay was used to quantify the biofilm metabolic activity; and the viable cell counting method was realized to determine cell viability and to discriminate the two species within the dual biofilm.

To complement data on anti-biofilm activity performance and provide additional information about the safety of BS coating for dental implantology applications, the impact of the selected biosurfactant on human primary osteoblasts was also addressed performing cytotoxicity test using both BS known concentrations and eluted solution from coated titanium discs.

The procedures for BS production, biofilm formation and quantification, evaluation of coating anti-biofilm efficacy, and cytotoxicity are detailed below.

### 2.2. Microbial Strains and Growth Conditions

The rhamnolipids-producing strain *Pseudomonas aeruginosa* 89, isolated from a patient with cystic fibrosis [59], was stored at −80 °C in Tryptic Soy Broth (TSB) (Scharlab Italia, Milan, Italy) supplemented with 25% glycerol and grown on Tryptic Soy Agar (TSA).

Biofilm assays were performed using two reference strains, *Candida albicans* ATCC ^®^ 10231 and *Staphylococcus aureus* ATCC ^®^ 6538. *C. albicans* ATCC ^®^ 10231 was stored at −80 °C in Sabouraud Dextrose Broth (Sigma-Aldrich, Milan, Italy) supplemented with 25% glycerol and grown on Sabouraud Dextrose Agar. *S. aureus* ATCC ^®^ 6538 was stored at −80 °C in TSB supplemented with 25% glycerol and grown on TSA. All these strains were incubated at 37 °C for 20 h before each experimental session.

### 2.3. Biosurfactant Production 

The biosurfactant R89 was produced, extracted and chemically characterized as described by Ceresa et al. [59]. Briefly, a loop of an overnight culture of *P. aeruginosa* 89 was inoculated into 40 mL of Nutrient Broth II (Sifin Diagnostics GmbH, Berlin, Germany) and incubated at 37 °C for 4 h at 140 rpm. Afterward, 24 mL of the bacterial suspension were inoculated in 1.2 L of Siegmund–Wagner medium and incubated at 37 °C for five days at 120 rpm. The culture was centrifuged (Sorvall RC-5B Plus Superspeed Centrifuge, Fisher Scientific Italia, Milan, Italy) at 7000 rpm for 20 min to remove bacterial cells. The supernatant was acidified with 6 M H_2_SO_4_ at pH 2.2, stored overnight at 4 °C and extracted three times with ethyl acetate (Merck KGaA, Darmstadt, Germany). The organic phase was anhydrified and evaporated to dryness under vacuum conditions. The composition of the raw extract was confirmed by mass spectrometry analysis.

### 2.4. Medical-Grade Titanium Discs Preparation

Titanium alloy Ti6Al4V (medical-grade 5) discs (TDs) 10 mm in diameter and 2 mm in thickness were obtained from computer numerical control machining (CLC Scientific s.r.l., Vicenza, Italy). TDs were subsequently polished with increasing fine-grained silicon-carbide abrasive paper up to 4000 grit to obtain a mirror surface. Cleaning and disinfection of the discs were performed as indicated in Tambone et al. [60]. Briefly, TDs were cleaned by sonication for 15 min each in three consecutive solutions, acetone, 70% *v/v* ethanol in distilled water, and distilled water, to remove impurities and grinding residues. The discs were then disinfected by immersion for 24 h in 70% *v/v* ethanol in water and stored in these conditions until further use. TDs were dried under a laminar flow immediately before testing.

### 2.5. Surface Coating Process

The coating of TDs was obtained by physical adsorption of the biosurfactant at the titanium surface, following the procedure reported by Tambone et al. [60]. Briefly, TDs were placed in 24-well polystyrene plates (one disc for each well) and immersed in 1 mL of a freshly prepared solution of R89BS 4 mg/mL in sterile phosphate buffer saline (PBS). The plates were incubated at 37 °C for 24 h and agitated at 70 rpm using an orbital shaker. At the end of the immersion period, the discs were aseptically transferred to new 24-well polystyrene plates and dried under a laminar flow to set the BS-coating at the surface.

### 2.6. Biofilm Growth on Titanium Surface

*C. albicans* ATCC ^®^ 10231 and *S. aureus* ATCC ^®^ 6538 single- and dual-species suspensions were prepared in Yeast Nitrogen Base (Sigma-Aldrich) with 50 mM dextrose (Scharlab Italia), supplemented with 10% *v/v* of fetal bovine serum (Pan Biotech ^™^, Aidenbach, Germany) (YNBD + 10% FBS). The final cell concentration was adjusted to 1 × 10^5^ colony forming units per mL (CFU/mL) for *C. albicans* and to 1 × 10^7^ CFU/mL for *S. aureus*.

Titanium discs (both coated and uncoated) were placed in sterile 24-well polystyrene plates, fitting one disc for each well. One milliliter of single- or dual-species suspension was added to each well, ensuring the complete submersion of the disc. Plates were then incubated at 37 °C in static condition for 1.5 h, to allow cells adhesion to the disc surface, and subsequently at 70 rpm up to 72 h. Every 24 h, discs were aseptically transferred into a new plate containing 1 mL of YNBD + 10% FBS, to provide fresh nutrients for the sessile cells.

At the end of the incubation period, the suspension was removed, and the discs were gently washed twice with sterile PBS to remove non-adherent cells.

### 2.7. Quantitative Tests for Biofilm Formation

To quantify single- and dual-species biofilm formation on TDs at the desired time-points three complementary methods were used: CV staining, MTT reduction assay and viable cell counting.

The crystal-violet method was performed to determine total biofilm biomass. Biofilms were air-dried at the surface of the disc and then stained with 1 mL of 0.2% *w/v* CV solution for 10 min. A set of TDs coated with the BS solution, but not incubated with the microbial suspension, was used as blank. After removing the CV solution, the discs were washed with distilled water to remove dye excess and air-dried again. Finally, the CV bound to the biofilms was dissolved in 1 mL of 33% *v/v* acetic acid (Scharlab Italia S.r.l., Milano, Italy) in water.

The MTT reduction assay was carried out to evaluate cell metabolic activity. An MTT working solution was prepared immediately before use dissolving 7.5 mg of MTT powder (Fisher Scientific Italia, Milan, Italy), 50 µL of glucose solution (20% *w/v* in distilled water) (Scharlab Italia S.r.l., Milano, Italy) and 100 µL of 1 mM menadione solution (Sigma-Aldrich Italia S.r.l., Milano, Italy) in 9.85 mL of PBS. Each disc was immersed in 1 mL of the 0.075% *w/v* MTT working solution and incubated for 30 min at 37 °C. R89BS-coated and uncoated discs without biofilm were also included in each MTT session as blanks. Finally, 1 mL of a lysing solution, composed of 7 parts of dimethyl sulfoxide (Scharlab Italia S.r.l., Milano, Italy) and 1 part of 0.1 M glycine buffer (pH 10.2) (Sigma-Aldrich Italia S.r.l., Milano, Italy), was used to dissolve the formazan crystals. The absorbance of CV and MTT resulting solutions was spectrophotometrically read at 570 nm (Victor^3^V ^TM^, Perkin Elmer, Milano, Italy).

The viable cell counting method was implemented to obtain information about dual-species biofilm composition and to evaluate the effect of R89BS coating on the single species within the multi-species biofilm. Each disc was placed in 50 mL tubes containing 10 mL of sterile NaCl solution (0.9% *w/v* in water) (Sigma-Aldrich Italia S.r.l., Milano, Italy) and subjected to four cycles of sonication and stirring, for 30 s each, to promote the detachment of cells from the titanium surface. The suspensions obtained were serially diluted 1:10 *v/v* in sterile NaCl solution. An aliquot of 100 µL of each multi-species biofilm dilutions was plated onto two selective agar media, Mannitol Salt Agar (MSA) (Scharlab Italia S.r.l., Milano, Italy), which is selective for *Staphylococcus* growth, and Sabouraud Chloramphenicol Agar (SCA) (Scharlab Italia S.r.l., Milano, Italy), selective for fungal growth. In addition, *S. aureus* and *C. albicans* single-species biofilm dilutions were plated onto MSA and SCA, respectively. Agar plates were incubated at 37 °C for 24 h and colonies were manually enumerated. All assays were performed in quadruplicate. CV tests were repeated three times, MTT tests were repeated two times, and viable cell count was performed on a single experimental session.

To understand the type of interaction established between the two microorganisms grown together, data obtained from the multi-species culture were compared to those obtained from both the single-species. In addition, data from the dual biofilm were also compared with the sum of data related to the two species. In the absence of interaction, the two microorganisms form a multi-species biofilm whose characteristics (biomass, metabolic activity and number of viable cells) should correspond to the sum of their respective single-species biofilms [61]. Differently, when the two species cooperate and promote each other’s growth, dual-species biofilm is expected to be higher than the sum of the individual biofilms. On the contrary, if competition occurs (e.g., when the two microorganisms contend for the nutrients), the resulting dual-species biofilm is expected to be lower than the sum of the single-species biofilms.

### 2.8. Scanning Electron Microscopy of Multi-Species Biofilms

In the second part of the study, in addition to the quantitative test reported above, a qualitative micro-morphological analysis of the multi-species biofilm formed on TDs at the different time-points was also carried out by scanning electron microscopy (SEM), using the sample preparation and imaging protocol described in Ceresa et al. [60,62] with minor modifications. Biofilms, formed on coated and uncoated controls discs, were immersed in 1 mL of 2.5% *w/v* glutaraldehyde solution (Scharlab Italia S.r.l., Milano, Italy) in 0.1 M phosphate buffer for 24 h at 4 °C to preserve the microstructural architecture of the biofilm on the titanium surface. Then, each disc was washed twice with Milli-Q ^®^ water, dehydrated by immersion in 70%, 90% and 100% *v/v* ethanol/water solutions for 10 min each and finally dried overnight under a laminar flow cabinet. Dried samples were then coated by a 10-nm layer of gold using a sputter coater (Emitech K500X, Quorum Technologies, Laughton, UK) to improve their electrical conductivity and thermal stability. 

SEM observation was performed using a Quanta 200 (FEI-Philips, Eindhoven, The Netherlands) scanning electron microscope in the high-vacuum mode. A set of four images for each disc were obtained by collecting the secondary electron signal at a magnification of 500×, 1000×, 2000× and 4000× to detect both the titanium surface morphology and the fine structural detail of both microbial cell species within the biofilm. The primary beam energy was set to 5 keV to minimize the damage to the biological structures. Possible artifacts due to the sample preparation process [63] were considered according to indications provided by Hrubanova et al. [64] and previous experience performed in imaging microbial biofilm formed in-vitro on medical devices [65,66,67] and in-vivo on titanium abutments [68,69].

### 2.9. Eukaryotic Cell Viability Tests

Human primary osteoblasts (hOBs) were obtained from human bone trabecular fragments provided by the Orthopaedic and Traumatology Unit, “Maggiore della Carità” Hospital, Novara, Italy. Written informed consent, specifying that residual material destined to be disposed of could be used for research, was signed by each participant before the biological materials were removed, in agreement with Rec(2006)4 of the Committee of Ministers Council of Europe on research on biological materials of human origin.

Bone fragments were washed with PBS and then digested with collagenase/elastase as described in Bosetti et al. [70]. The isolated bone outgrowths of cells from the digested bone fragments appeared in culture dishes within one week and formed a confluent monolayer at 3–4 weeks. The cells were characterized including osteoblastic morphology, alkaline phosphatase expression and hormone responsiveness (PTH, 1.25(OH)-D_3_) [71]. Cells within the eighth passage were maintained in Iscove’s modified Dulbecco’s medium (IMDM, Euroclone, Milano, Italy) supplemented with 10% fetal bovine serum (FBS, Hyclone GE Healthcare, Logan, UT, USA), penicillin/streptomycin (PS, 50 U/mL and 15 μg/mL) and 2 mM L-glutamine (Glu) at 37 °C in 5% CO_2_. Where not specified, reagents were from Sigma-Aldrich Italia S.r.l. (Milano, Italy). Osteoblasts were plated in a 48 microplate well at a concentration of 5 × 10^3^ cells per well. 

Cytotoxicity on hOBs was evaluated both on pre-defined concentrations of R89BS in the growth medium and using eluates from titanium discs coated with R89BS, as previously described in [60]. R89BS pre-defined concentrations ranged from 0 to 200 μg/mL and their effect on cells was measured after 24 h. For cytotoxicity test of R89BS released from titanium discs, hOBs were cultured for 24, 48 and 72 h in 500 μL of the conditioned surface-contacting medium (CS-CM) obtained incubating titanium discs uncoated or coated with R89BS in culture media for 24 h at 37 °C.

Cell viability was evaluated using MTT as an indicator of cell viability. MTT test was done by adding 50 µL of MTT solution (5 mg/mL) to the cell monolayers and then incubated for 3 h at 37 °C in the dark. After culture media removal and PBS washing, formazan crystals formed during the process were extracted by adding 200 μL of HCl/isopropanol solution prepared by adding 250 μL of 1 M hydrochloric acid in 5 mL of isopropanol. After 20 min, the absorbance was measured at 570 nm with a spectrophotometer (VICTOR^3^V ^TM^, PerkinElmer, Inc., Waltham, MA, USA).

Experiments were carried out in triplicate and repeated three times. Data were expressed as % vitality considering 100% vitality the values obtained from cells treated with no R89BS addition. As cytotoxicity control, cells treated with 1% (*v/v*) Triton-X 100 were used.

In accordance with ISO 10993-5:2009, cytotoxicity was defined when viability was <70% of the untreated controls.

### 2.10. Data Analysis and Statistics

The single TD was considered as a statistical unit. Quantitative data obtained from replicated CV and MTT tests were normalized to the value of the corresponding blank and were expressed as mean values of absorbance and standard deviation. The inhibition percentages of biofilm formation were calculated using the following formula:$$Inhibition\;\left(\%\right)=\left(1-\frac{A_{R89BS}}{A_{Ctrl}}\right)\times 100$$
where *A_R89BS_* is the absorbance value of BS-coated samples and *A_Ctrl_* is the absorbance value of the untreated control.

Data obtained from viable cell counting were reported as the mean value of CFU/disc. The inhibition percentages of biofilm formation were determined as follows:$$Inhibition\;\left(\%\right)=\left(1-\frac{T}{Ctrl}\right)\times 100$$
where *T* is the CFU mean value of BS-coated discs and *Ctrl* is the CFU mean value of the uncoated control discs.

The percentages of eukaryotic cell viability were estimated using the following formula:$$Cell\;viability\;\left(\%\right)=\frac{A_{sample}}{A_{Ctrl}}\times 100$$
where *A_sample_* is the absorbance value of sample treated with R89BS or Triton-X 100 and *A_Ctr_*_l_ is the absorbance mean value of untreated controls. 

Statistical analysis was elaborated using the statistical program R, 3.5.3 (R Development Core Team, http://www.R-project.org, accessed on 1 May 2021). In the biofilm formation study, one-way ANOVA followed by Tukey post-hoc test was used to compare single- and multi-species biofilm, whilst the sum of the two single-species biofilms and the multi-species biofilm were compared using Welch two-sample *t*-test. The effect of R89BS-coated TDs on *C. albicans-S. aureus* multi-species biofilm formation was investigated with Student’s *t*-test comparing coated discs with uncoated controls at each time-point. One-way ANOVA followed by Tukey post-hoc test was used to study the significance of data in the cytotoxicity assay in comparison to positive and negative controls. Results were considered statistically significant when *p* < 0.05.

## 3. Results

### 3.1. Formation of C. Albicans and S. Aureus Single- and Dual-Species Biofilm on Titanium Discs

The ability of *C. albicans* ATCC ^®^ 10231 and *S. aureus* ATCC ^®^ 6538 to develop biofilm alone and in combination on untreated titanium discs was investigated up to 72 h of incubation, using a range of different methods.

The designed experimental conditions allowed the growth of *C. albicans* and *S. aureus* and promoted the development of both the single-species biofilms. However, *C. albicans* biofilm amount, in terms of biofilm biomass, cell metabolic activity and the number of viable cells, resulted significantly higher than *S. aureus* (*p* < 0.001) formed biomass at all time-points. When the two strains were co-cultured, they formed a thick and dense dual-species mixed biofilm on the titanium surface. Both, mono- and multi-species biofilms, gradually increased over time, in terms of biomass, metabolic activity and viable cell counts (Figure 1). The inspection of data obtained with CV method, MTT assay and viable cell counting (Figure 1a–c) shows that in all but two cases (metabolic activity at 24 h and viable cell counting at 72 h), multi-species biofilm amount resulted significantly lower than the sum of the individual biofilms (*p* < 0.001), pointing out a competitive interaction between *C. albicans* and *S. aureus* within the biofilm model of this study.

Viable cell counting provided additional insight into the characteristics of *C. albicans* and *S. aureus* mixed biofilm with time. CV and MTT assay allow an overall evaluation of multi-species biofilm, but no information can be inferred about the single species that compose it. For example, despite an overall reduction of multi-species biofilm compared to the sum of the individual biofilms, one species can benefit from the co-culture while the other is damaged. To investigate how the two microorganisms affected each other in the biofilm formation process, the number of viable cells of each species within multi-species biofilms was compared to that obtained when grown on their own.

In the co-culture, a reduction in the number of viable cells of both strains was observed. The number of CFU/disc of *C. albicans* in the multi-species biofilm resulted significantly lower than that of the single culture at 24 and 48 h, with an overall reduction of 36% (*p* < 0.01). *S. aureus*, instead, was significantly reduced at all the time-points, with a reduction of 59%, 73% and 76% after 24, 48 and 72 h respectively (*p* < 0.001).

The composition of the multi-species biofilm was finally determined on MSA, selective for *Staphylococcus,* and on SCA, selective for fungal growth. The percentages of *C. albicans* and *S. aureus* at the different time-points are shown in Figure 1d. Multi-species biofilm results were mainly composed of *C. albicans* cells while the percentage of *S. aureus* was approximately halved every 24 h, resulting in 3% after 72 h of incubation. *C. albicans*, therefore, resulted the best biofilm producer, not only in a single culture but also when co-incubated with the bacterial strain.

### 3.2. Anti-Biofilm Activity of R89BS-Coated Titanium Discs against Multi-Species Biofilm

The ability of TDs coated with 4 mg/mL R89BS solution to inhibit *C. albicans*-*S. aureus* multi-species biofilm formation was evaluated by CV method, MTT reduction assay and viable cell counting performed at 24, 48 and 72 h of incubation.

As observed in the previous phase of the study, multi-species biofilm developed on uncoated control titanium discs gradually increased over time, in terms of biomass, metabolic activity and cell viability, and a similar trend was observed for the biofilm formed on R89BS-coated TDs (Figure 2). However, comparing the two surfaces (coated and uncoated), significantly lower values were observed on coated TDs for biofilm biomass, cell metabolic activity and the number of viable cells up to 48 h of incubation. R89BS coating resulted more effective in inhibiting biofilm biomass (Figure 2a) rather than cell metabolic activity and cell viability (Figure 2b,c).

From a quantitative point of view, the highest ability of R89BS-coated TDs to reduce biofilm formation was shown at 24 h, with an inhibition of the biofilm biomass, cell metabolic activity, and cell viability higher than 90% (*p* < 0.001). After 48 h, the inhibition of biofilm biomass and cell viability reached 36% (*p* < 0.001) and 29% (*p* < 0.01), respectively, while a less marked effect was observed for metabolic activity, with an inhibition of 14% (*p* < 0.01). No significant difference was observed comparing the values obtained for biofilm grown on control and coated discs after 72 h of incubation. Table 1 summarizes the percentages of biomass, metabolic activity and cell viability inhibition at the different time-points.

To evaluate whether the R89BS coating was able to contrast mainly one of the two species within the multi-species biofilm, the number of cells of *C. albicans* and *S. aureus* were determined for both coated and uncoated discs. At 24 h, R89BS coating resulted more effective in reducing the number of fungal cells, with an inhibition of 92% for *C. albicans* and 81% for *S. aureus*. In contrast, after 48 h of incubation *S. aureus* was the species showing the highest inhibition, with a reduction in the cells number of 69% for the bacterial strain and of 27% for the fungal strain. At 72 h, the cells number of each species on R89BS-coated TDs was comparable to that on uncoated control discs (*p* > 0.05).

Using selective media (MSA for *Staphylococcus,* and SCA for *Candida*) a higher prevalence of fungal cells was observed on coated discs, at all the time-points, as shown in Figure 2d. The composition of the biofilm on control discs resulted the same as observed in the first phase of the study (Figure 1d). Interestingly, at 24 h, the R89BS coating of the titanium surface was associated with a doubling of *S. aureus* percentage compared to that observed for the controls. At prolonged incubation times, the percentage of the bacterial strains was significantly reduced and remained constant between 48 and 72 h.

Images obtained by SEM inspection of the biofilm grown on R89BS coated and untreated TDs are presented in Figure 3.

Both microbial species were present at all time-points in both treated and untreated samples, confirming quantitative data presented in Figure 2d. A clear difference in the amount of both species at the titanium surface was evident at 24 h. The comparison of treated and untreated samples at 48 h of incubation showed differences in the three-dimensional development of the biofilm, being thicker on uncoated than coated samples. No major differences were found in the amount of titanium cell-free surface, being the samples mostly covered by both microorganisms irrespectively from the treatment.

Coccoid cells were found prevalently at the titanium surface or along with *Candida* hyphal bodies. Interestingly, bacterial cells appeared to prefer deeper layers in the three-dimensional structured biofilm imaged at 48 and 72 h.

### 3.3. R89BS Effect on Eucaryotic Cell Viability

Results of R89BS activity on hOBs viability are shown in Figure 4. A reduction of hOBs cell viability below 70% occurred for R89BS concentrations higher than 50 μg/mL (Figure 4a).

No interference with hOBs growth was found for concentrations equal and lower than 50 μg/mL, resulting in cell viability >80%. Furthermore, no cytotoxic effect was observed on hOBs cultured in the eluate from the R89BS-coated titanium. MTT data were comparable to those obtained from cells grown in the eluate of untreated titanium discs at 24 h. The extension of the experiment to longer time-points (48 and 72 h) showed no decrease in cell viability as reported in Figure 4b, showing data obtained at 72 h of incubation.

## 4. Discussion

Limited data are currently available in the literature about biosurfactants activity against mixed cultures of microorganisms, although polymicrobial infections are held responsible for several human diseases, including infections of skin, urinary tract and oral cavity, otitis media, foot wound infections in diabetic patients and chronic infection in the cystic fibrosis lung [16,72]. Zezzi do Valle Gomes and Nitschke [73] observed that a polystyrene surface pre-conditioned with 1% aqueous solution of rhamnolipid produced by *P. aeruginosa* LBI, reduced by 44.5% the adhesion of a mixed culture of the two food-borne pathogenic bacteria *S. aureus* and *Listeria monocytogenes.* Dìaz De Rienzo and collaborators [74] reported the ability of a 5% *v/v* solution of sophorolipid S1, produced by *C. bombicola* ATCC ^®^ 22214, to disrupt pre-formed biofilms of *S. aureus* and *Bacillus subtilis*.

Recently, the authors observed that the rhamnolipid R89 used in this study has remarkable antibacterial and anti-biofilm properties against *Staphylococcus* spp. The raw extract of R89BS is a mixture of homologues of mono- (75%) and di-rhamnolipids (25%), as detected by mass spectrometry [59]. R89BS has proven to limit the biofilm formation of single cultures of *S. aureus* and *Staphylococcus epidermidis* when used as a coating agent of different medical-grade surfaces, like silicone [59] and titanium [60], demonstrating the potential applicability of this molecule in preventing the colonization of indwelling and implantable medical devices. Furthermore, in a recent work [75], R89BS also showed significant anti-biofilm properties against polymicrobial cultures of *C. albicans* and *Staphylococcus* spp. Polystyrene and medical-grade silicone surfaces were coated with R89BS, and their anti-biofilm efficacy was evaluated against mixed cultures of *C. albicans-S. aureus* and *C. albicans-S. epidermidis* up to 72 h, resulting in an overall inhibition greater than 90% in both cases.

In this study, the rhamnolipids-coating was realized by R89BS physical adsorption at the titanium surface, and its anti-biofilm efficacy was investigated on *C. albicans* and *S. aureus* dual-species culture up to three days. Among fungal-bacterial cultures, the binomial *C. albicans*-*S. aureus* is among the most studied, as both microorganisms are often co-isolated in several pathologies, including oral diseases like periodontitis, denture stomatitis and medical devices related infections [16,30,76]. Enteric rods were recovered more frequently and at higher levels in peri-implantitis compared with periodontitis [77], and *P. aeruginosa*, *S. aureus* and *C. albicans* were frequent in peri-implantitis, suggesting they may be associated with implant failure [78]. These microorganisms may co-exist, cooperating to increase mutual resistance to antimicrobial treatments and immune system, representing a significant therapeutic challenge [16,19,79].

We showed the ability of *C. albicans* and *S. aureus* to co-colonize the surface of medical-grade titanium typically used for both dental implants and trans-mucosal abutments. The designed experimental model allowed the reproducible formation of a thick and well-adherent multi-species biofilm up to 72 h, composed mainly of fungal cells. The comparison between single- and multi-species biofilms revealed that an antagonistic relationship was established between *C. albicans* and *S. aureus*, in which both species, and especially *S. aureus*, were impacted by the co-incubation. Shirtliff et al. [19] proposed that at some point during the biofilm formation process, the initially synergistic interaction between *C. albicans* and *S. aureus* may turn to a competitive or antagonistic relationship. A possible explanation for this type of interaction is that microorganisms compete for the available nutrients [80]. In addition, both species showed an affinity for the titanium surface when grown individually, forming a stable and solid biofilm, up to 72 h. Therefore, a further possible explanation for the antagonistic interaction is that the two species compete for the available adhesion sites, preventing in this way the other species adhesion to the surface, with a mechanism called “surface blanketing” [81]. Although the mechanisms behind microbial interactions have not been fully elucidated, different solutions have been proposed to explain the antagonism between the different species. Microorganisms can produce a wide range of secondary metabolites which may be toxic to other species or may interfere with biofilm development, hampering initial adhesion or cell to cell communication, inducing matrix degradation or biofilm dispersion [81]. For example, it was observed that farnesol, a quorum-sensing molecule produced by *Candida*, compromises *S. aureus* biofilm formation, and has a toxic effect on cells and increase susceptibility to several antimicrobial agents [82]. In contrast, other authors reported that *C. albicans* and *S. aureus* established a synergistic relationship with a significant increase of biofilm biomass when co-incubated [75,83,84]. However, the interactions between the different microorganisms are complex and strongly dependent on the context. In an in vitro model, for example, several factors influence microbial interaction, such as the medium, the surface, cell concentration, and when microorganisms are added to the culture (simultaneously or at different times during biofilm development) [85].

The designed model of *C. albicans*-*S. aureus* dual-species biofilm was then exploited to evaluate the anti-biofilm activity of a titanium surface coated with R89BS. Rhamnolipids-coating was able to significantly reduce microbial adhesion over time, showing a remarkable effect at 24 h with a biofilm inhibition of about 94%. The inhibition efficacy decreased with time but was still able to guarantee a significantly lower amount of biofilm in coated samples compared to uncoated controls at 48 h. Both species within the mixed culture, but mostly *S. aureus*, were affected by R89BS. In this regard, it should be considered that the anti-biofilm activity observed against *S. aureus* is related to a double action of the biosurfactants, being both anti-adhesive and antibacterial in respect to this species. *S. aureus*, in fact, showed a significant susceptibility to R89BS, with a minimal inhibitory concentration (MIC) of 0.06 mg/mL, as reported in previous work [59]. On the contrary, R89BS showed only an anti-adhesive activity against *C. albicans*, since no antimicrobial activity was observed against the fungus at the tested concentrations (unpublished data).

The anti-adhesive activity of biosurfactants can be related to their ability to modify the properties of both coated surfaces and cellular membranes of the microorganisms they come into contact with. It is known that the BS adsorbed on a surface forms a thin layer which, thanks to the orientation of the BS amphiphilic molecules, reduces its hydrophobicity and surface tension and increases its wettability [86]. It has also been proposed that BS may alter microbial cell membrane hydrophobicity and electric charges [87,88]. Thanks to these properties, BS can interfere with cell–cell and cell–surface interactions, reducing the hydrophobic interactions responsible for the initial adhesion of the microorganism to the surface. Moreover, microbial adhesion can be hindered by the electrostatic repulsion established between the negative charges of the anionic rhamnolipid and the negative charges on the microbial surface [73,89].

The spatial organization of *C. albicans* and *S. aureus* within the dual-species biofilm was observed at SEM, confirming quantitative data. In agreement with our findings, other authors have reported that, in a dual-species biofilm, *C. albicans* and *S. aureus* are in close contact with each other, with bacterial cells preferentially attached to the fungal hyphae [90], and a key role in this interaction is played by the fungal adhesine Als3p [91]. This affinity of *S. aureus* for *C. albicans* hyphae could play a crucial role in the increased virulence of infection sustained by this fungal-bacterial biofilm. In fact, the ability of *C. albicans* hyphae to penetrate through the epithelium and mucous membranes [92] can lead to the dissemination of *S. aureus* from the oral cavity to the bloodstream, resulting in systemic infection as observed by Schlecht et al. [93] in immunocompromised mice with an oral co-infection of these two microorganisms. 

A relevant aspect that makes BS good candidates in medical applications is their biocompatibility due to their low toxicity towards eukaryotic cells. Previous studies on medical-grade silicone coated with R89BS assessed in-vitro cytotoxicity of R89BS using immortalized human lung fibroblasts (MRC5) [59]. The biocompatibility of titanium discs coated with R89BS was also assessed, testing the release of the molecule from the coated titanium implant into the aqueous environment. No cytotoxic effect was found against MRC5 cell line when exposed to the R89BS-coated titanium discs eluate [60]. In agreement with the previously reported data, the results of the cell viability test performed in this study showed no change in the viability of primary osteoblasts when put in contact with the eluate of the R89BS-coated titanium. Of note, these data were experimentally confirmed also at 48 and 72 h of incubation, a time markedly longer than that usually recommended in the standards.

Although the reported results are relevant for defining new anti-biofilm strategies for dental implant applications, some study limitations should be considered. The two selected microorganisms are widely recognized as biofilm former strains, but further tests must be performed, possibly including biofilm-forming strains of relevance for the peri-implant diseases [94]. In addition, the currently available knowledge of the peri-implant microbiome can also indicate the most appropriate strains to be considered for further tests [10,95]. Major limitations occur, however, in realizing reliable biofilm models in-vitro due to complex culturing conditions required by most of the oral microorganisms. Better testing conditions could be possibly presented by animal in-vivo studies, which should be considered in the future.

A second technical limitation is the progressive loss of the anti-biofilm efficacy with time. The anti-biofilm activity reported in this study showed significant effects limited to 48 h. This short-term effect is relevant to protect the implant after surgical placement, but extended efficacy is needed to prevent the onset of the peri-implant diseases at later stages after installation. Alternative bonding strategies, strengthening the coating adhesion or limiting the anti-biofilm molecule released into the aqueous environment should be considered to improve the durability of R89BS-coating and its anti-biofilm efficacy. However, the development of alternative strategies is complicated by the need to preserve implant biocompatibility.

## 5. Conclusions

In conclusion, considering the anti-biofilm and the toxicity test results, a satisfactory balance was obtained in-vitro by the use of R89BS-coated titanium, preventing on the one hand a good anti-biofilm effect that could impact implant septic failure and, on the other hand, a minimal cytotoxic effect that should not impair osseointegration process. Overall, these data enlarge the body of evidence that supports further testing toward in-vivo applications.

Looking forward to the potential clinical applicability of the coating in contrasting biofilm-related implant diseases, several different ways of use can be identified, according to where and when the coating is applied. Among possible applicative scenarios, the R89BS coating could be potentially applied on the implant itself, or in its most coronal portion, or could be used to coat the transmucosal prosthetic components, where microbial colonization is early and massive. As for the timing, it could be applied during the industrial production of the implants and related components or considered for a local application by the dentist during the implant positioning or maintenance therapy.

## Figures and Tables

**Figure 1 polymers-13-02420-f001:**
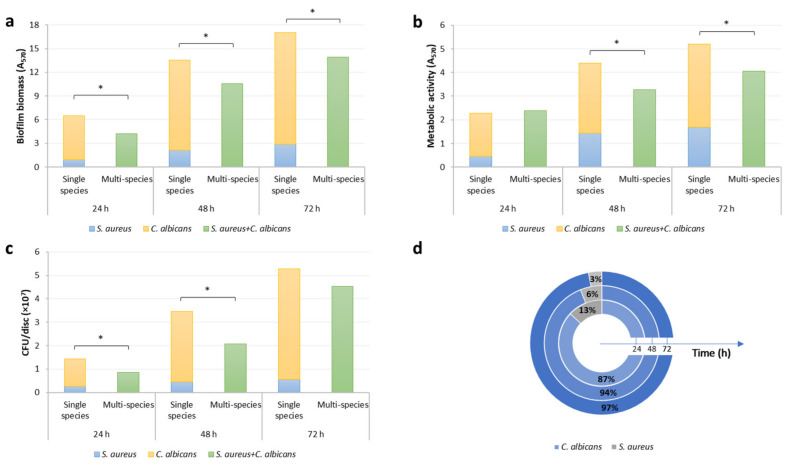
Biomass (**a**), metabolic activity (**b**) and cell viability (**c**) of *C. albicans* ATCC ^®®^ 10231 and *S. aureus* ATCC ^®®^ 6538 single- and multi-species biofilm formed on untreated titanium discs. Results were obtained by mean CV staining, MTT reduction assay and viable cell counting respectively. Single- and multi-species cultures were grown in YNBD + 10% FBS and were evaluated after 24, 48 and 72 h of incubation. The multi-species biofilm composition (**d**) was determined by viable cell counting on selective agar media (Mannitol Salt Agar, selective for *Staphylococcus,* and Sabouraud Chloramphenicol Agar, selective for fungal growth) at the same time-points (24 h: internal pie; 48 h: middle pie; 72 h: external pie). * *p* < 0.05.

**Figure 2 polymers-13-02420-f002:**
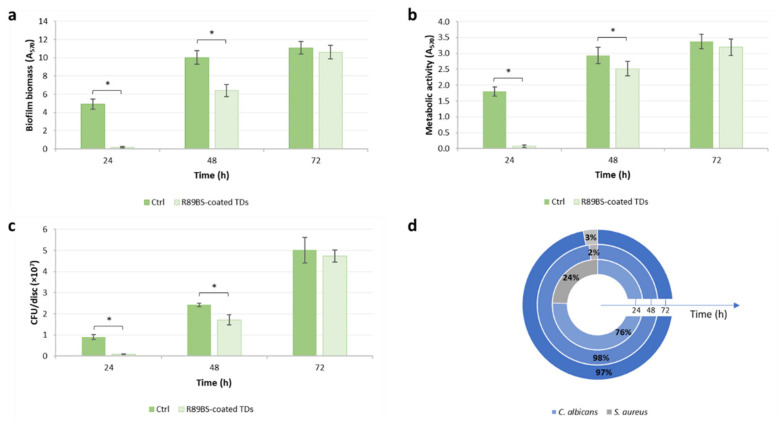
Biomass (**a**), metabolic activity (**b**) and cell viability (**c**) of *C. albicans*-*S. aureus* multi-species biofilm formed on untreated control TDs (dark bars) and TDs coated with 4 mg/mL R89BS (light bars). Results were obtained by means of CV staining, MTT reduction assay and viable cell counting respectively, after 24, 48 and 72 h of incubation. The composition of the multi-species biofilm formed on R89BS-coated TDs (**d**) was determined by viable cell counting on selective agar media (Mannitol Salt Agar, selective for *Staphylococcus,* and Sabouraud Chloramphenicol Agar, selective for fungal growth) at the same time-points (24 h: internal pie; 48 h: middle pie; 72 h: external pie). Error bars represent standard deviation. * *p* < 0.05.

**Figure 3 polymers-13-02420-f003:**
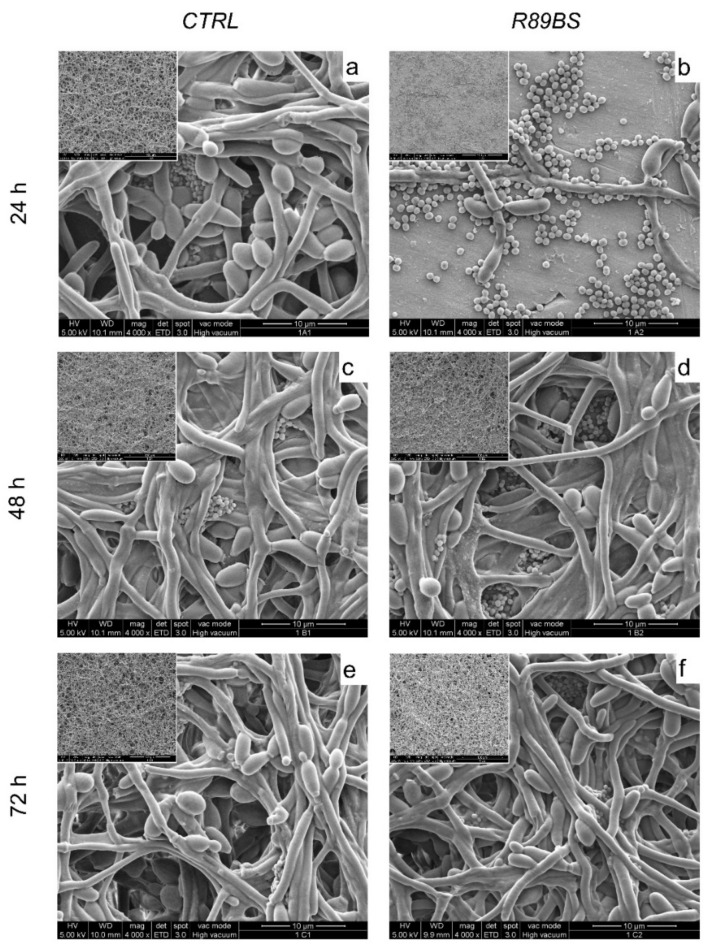
Architecture of *C. albicans-S. aureus* dual-species biofilm grown on uncoated control TDs (*CTRL*, panels (**a**,**c**,**e**)) and R89BS-coated TDs (*R89BS*, panels (**b**,**d**,**f**), after 24, 48 and 72 h of incubation. Representative images were obtained by scanning electron microscopy in high-vacuum mode. Areas free of adherent microorganisms in (**b**) are well representing the micro-morphology of both coated and uncoated titanium discs. Original magnification: 4000× (insets 500×).

**Figure 4 polymers-13-02420-f004:**
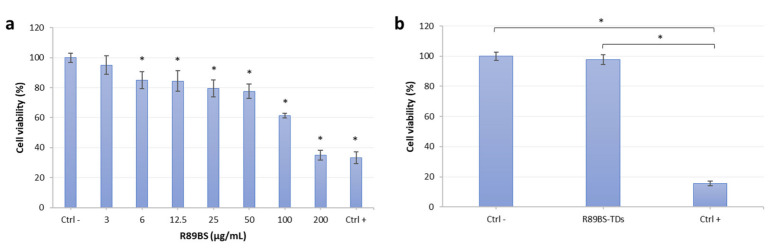
Results of the eucaryotic cell viability MTT test: (**a**) hOBs viability data with R89BS added as aqueous solution at concentrations ranging from 200 to 0 µg/mL (Ctrl −); (**b**) Viability data obtained from hOBs cultured in the eluate from R89BS-coated discs (R89BS-TDs) and uncoated titanium discs (Ctrl −). Triton X-100 was used as positive control (Ctrl +). Error bars represent standard deviations. (* *p* < 0.05).

**Table 1 polymers-13-02420-t001:** Inhibition percentages of *C. albicans-S. aureus* multi-species biofilm determined by CV staining (biomass), MTT reduction assay (metabolic activity) and viable cell counting (cell viability), after 24, 48 and 72 h. Data are reported as mean and standard deviation.

Incubation Time (h)	Inhibition (%)		
	Biomass	Metabolic Activity	Cell Viability
24	96.3 (1.2) *	95.9 (1.8) *	90.6 (1.1) *
48	36.1 (6.7) *	14.0 (7.7) *	29.1 (9.8) *
72	4.4 (6.7)	5.3 (7.8)	5.4 (4.5)

* *p* < 0.05.

## Data Availability

The datasets used and analyzed during the current study are available from the corresponding author on reasonable request.

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
