# Peer review of "Counter-Acting Candida albicans-Staphylococcus aureus Mixed Biofilm on Titanium Implants Using Microbial Biosurfactants"

_polymers, 2021, doi:10.3390/polym13152420_

Round 1

Reviewer 1 Report

Dear Authors

Title of the study is too long. I would suggest revising it. 

Abstract: it is well written

Keywords: Need to add more appropriate which reflect the below work. 

Introduction:

Line 36: this statement needs a suitable reference. this reference information also good for introductory line;

-Khurshid, Z., Naseem, M., Sheikh, Z., Najeeb, S., Shahab, S., & Zafar, M. S. (2016). Oral antimicrobial peptides: Types and role in the oral cavity. Saudi Pharmaceutical Journal24(5), 515-524.

after line- 48: Authors can write two or three lines on titanium property and little introduction of dental implant here. the below reference is good;

  • Najeeb, Shariq, et al. "Dental implants materials and surface treatments." Advanced Dental Biomaterials. Woodhead Publishing, 2019. 581-598.

In study design heading: Authors not discussed from where they get an idea of reporting the experiment. Or this is the first time they doing it? I mean they get idea of this work from some past reported paper. please cite them,

In 2.6 heading: Authors have to make sure about company trademarks and name properly. 

Figure-1 & 2 resolution is not well. 

Is any limitations faced during this original work? if yes disclose them so further research will be better. 

Author Response

Ref.:  Ms. No. polymers-1300571

Response to Reviewer # 1

We like to thank the reviewer for the careful evaluation of our manuscript and for the useful comments, which have helped to improve our manuscript.

Below we respond point by point to the various comments and indicate the changes we have made in the revised manuscript. English has been carefully checked throughout the manuscript. Revised text is highlighted in the new manuscript version.

Q: Title of the study is too long. I would suggest revising it.

A: Title has been shortened, according to the reviewer suggestion.

Q: Abstract: it is well written

A: We thank the reviewer for this positive feedback.

Q: Keywords: Need to add more appropriate which reflect the below work.

A: The keywords list has been expanded, integrating it with other relevant elements.

Q: Introduction: Line 36: this statement needs a suitable reference. this reference information also good for introductory line; -Khurshid, Z., Naseem, M., Sheikh, Z., Najeeb, S., Shahab, S., & Zafar, M. S. (2016). Oral antimicrobial peptides: Types and role in the oral cavity. Saudi Pharmaceutical Journal, 24(5), 515-524.

A: References has been added as suggested (References 1 and 2 of the manuscript revised version). See line 37 of the revised manuscript.

Q: After line- 48: Authors can write two or three lines on titanium property and little introduction of dental implant here. the below reference is good; Najeeb, Shariq, et al. "Dental implants materials and surface treatments." Advanced Dental Biomaterials. Woodhead Publishing, 2019. 581-598.

A: A short description of the titanium properties was added, including relevant references (References 11-14 of the manuscript revised version). See lines 46-49 of the revised manuscript.

Q: In study design heading: Authors not discussed from where they get an idea of reporting the experiment. Or this is the first time they doing it? I mean they get idea of this work from some past reported paper. please cite them

A: A short clarification about the genesis of the study and the previous findings was added in the “Study design” heading, including the citation of the paper showing the ability of biosurfactant coating to inhibit the formation of single Staphylococcus spp. biofilm on titanium (Reference 60 of the manuscript revised version). See lines 109-111 of the revised manuscript.

Q: In 2.6 heading: Authors have to make sure about company trademarks and name properly.

A: Trademarks and registered marks were added when appropriate. See lines 178-180 of the revised manuscript.

Q: Figure-1 & 2 resolution is not well.

A: We apologized for the inconvenience, possibly due to pdf conversion of the manuscript file. High resolution figures (600 dpi) were sent to the editor separately from the manuscript file in order to assure figure quality in case the manuscript is accepted for publication.

Q: Is any limitations faced during this original work? if yes disclose them so further research will be better.

A: Two additional paragraphs were added at the end of the Discussion section to address the main limitations of the study. See lines 571-589 of the revised manuscript.

Reviewer 2 Report

manuscript is well written and easy to follow the concept and goal of the research. I have a few question and suggestion.

  1. In line 425 you have used word "coated and untreated " either use coated/ uncoated or treated/untreated. 
  2. why did you choose the concentration (4mg/ml) of R89 BS to coat the TD. Does the  increase in concentration of R89BS will increase the amount of BS coating in TD?
  3.  Since you have prepared the R89BS by yourself , have you done any characterization or any other experiment to check its purity? you have mention that the raw extract was confirmed by mass spectrometry analysis but its result is not included in manuscript.
  4. in Figure 3 add SEM image of coated and uncoated TD without microorganism.
  5.  in line 214-217 you have mention the use of 2 selective media but for explanation of  figure 1(d) and 2(d) you have not discussed which media was used for which organism.  for different selective media the result may differ.

Author Response

Ref.:  Ms. No. polymers-1300571

Response to Reviewer # 2

We like to thank the reviewer for the positive evaluation of our manuscript and for his useful comments, which will help to improve our research activity in this field.

Below we respond point by point to the various comments and indicate the changes we have made in the revised manuscript. Revised text is highlighted in the new manuscript version.

Q: In line 425 you have used word "coated and untreated " either use coated/ uncoated or treated/untreated.

A: Amended as indicated. See line 436 of the revised manuscript.

Q: Why did you choose the concentration (4mg/ml) of R89 BS to coat the TD. Does the  increase in concentration of R89BS will increase the amount of BS coating in TD?

A: We thanks the reviewer for this relevant question. Indeed, in principle, the higher the concentration of the coating solution, the higher the amount of the BS coating on the titanium surface. The concentration of 4 mg/ml was applied in accordance with the results of a previous optimization study, comparing a range of different concentrations and showing that 4 mg/ml was the best concentration to guarantee a significant anti-biofilm activity without impacting on cytotoxicity of coated sample. These data were reported in a recently published paper (Tambone et al BMC Oral Health 2021, 21, 49, doi:10.1186/s12903-021-01412-7)

Q: Since you have prepared the R89BS by yourself , have you done any characterization or any other experiment to check its purity? you have mention that the raw extract was confirmed by mass spectrometry analysis but its result is not included in manuscript.

A: R89BS characterization has been presented before in the paper from Ceresa et al (Ceresa et al. Molecules 2019, 24, doi:10.3390/molecules24213843). In that paper, we reported that the negative electrospray ionization (ESI) MS analysis of the crude biosurfactant R89BS extract showed the presence of homologues of mono- and di-rhamnolipids. The mono-rhamnolipid family members were composed mainly of C10–C10, C8–C10, and C10–C12 homologues. The di-rhamnolipid family members were represented by three main homologues corresponding to C10–C8, C10–C10, and C10–C12 homologues. The relative amounts of the two groups in the crude extract were about 75% mono- and 25% di-rhamnolipids. Further details are available in the above-mentioned paper.

Q: In Figure 3 add SEM image of coated and uncoated TD without microorganism.

A: SEM images of titanium discs equivalent to those used in this study with and without R89BS coating were reported in a recent publication from our group (Tambone et al BMC Oral Health 2021, 21, 49, doi:10.1186/s12903-021-01412-7). Since no changes are visible at the SEM magnifications used in Figure 3 (and even at higher magnifications) we considered of no additional value to include those images as part of Figure 3. Areas free of adherent microorganisms in panel b of Figure 3 are well representing the micro-morphology of both coated and uncoated titanium discs. This information has been added to the Figure 3 legend. See lines 430-432 of the revised manuscript.

Q: In line 214-217 you have mention the use of 2 selective media but for explanation of figure 1(d) and 2(d) you have not discussed which media was used for which organism. For different selective media the result may differ.

A: For sake of clarity, the used selective media were further specified in the legends of Figure 1 (lines 352-353) and Figure 2 (lines 393-394) and within the text of both headings 3.1 (lines 366-369) and 3.2 (line 418).